# Transcriptome Metabolic Characterization of *Tuber borchii* SP1—A New Spanish Strain for In Vitro Studies of the Bianchetto Truffle

**DOI:** 10.3390/ijms241310981

**Published:** 2023-07-01

**Authors:** Emilia Chuina Tomazeli, Manuel Alfaro, Alessandra Zambonelli, Edurne Garde, Gumer Pérez, Idoia Jiménez, Lucía Ramírez, Hesham Salman, Antonio G. Pisabarro

**Affiliations:** 1Institute for Multidisciplinary Research in Applied Biology (IMAB), Public University of Navarra (UPNA), 31006 Pamplona, Spain; 2Bionanoplus, 31194 Oricain, Spain; 3Department of Agro-Food Sciences and Technologies, University of Bologna (UNIBO), 40126 Bologna, Italy

**Keywords:** axenic culture, aroma genes, central metabolism, fermentation, secondary metabolite clusters, strain isolation, trasncription effort, transcriptome profile, truffle cultivation, vegetative mycelium

## Abstract

Truffles are ascomycete hypogeous fungi belonging to the *Tuberaceae* family of the *Pezizales* order that grow in ectomycorrhizal symbiosis with tree roots, and they are known for their peculiar aromas and flavors. The axenic culture of truffle mycelium is problematic because it is not possible in many cases, and the growth rate is meager when it is possible. This limitation has prompted searching and characterizing new strains that can be handled in laboratory conditions for basic and applied studies. In this work, a new strain of *Tuber borchii* (strain SP1) was isolated and cultured, and its transcriptome was analyzed under different in vitro culture conditions. The results showed that the highest growth of *T. borchii* SP1 was obtained using maltose-enriched cultures made with soft-agar and in static submerged cultures made at 22 °C. We analyzed the transcriptome of this strain cultured in different media to establish a framework for future comparative studies, paying particular attention to the central metabolic pathways, principal secondary metabolite gene clusters, and the genes involved in producing volatile aromatic compounds (VOCs). The results showed a transcription signal for around 80% of the annotated genes. In contrast, most of the transcription effort was concentrated on a limited number of genes (20% of genes account for 80% of the transcription), and the transcription profile of the central metabolism genes was similar in the different conditions analyzed. The gene expression profile suggests that *T. borchii* uses fermentative rather than respiratory metabolism in these cultures, even in aerobic conditions. Finally, there was a reduced expression of genes belonging to secondary metabolite clusters, whereas there was a significative transcription of those involved in producing volatile aromatic compounds.

## 1. Introduction

Truffles are ascomycete hypogeous fungi belonging to the *Tuberaceae* family of the *Pezizales* order [1]. Although different genera of this order produce subterranean ascomata, several authors only consider “true truffles” to be the species belonging to the genus *Tuber* [2,3]. *Tuber* spp. grow in ectomycorrhizal symbiosis with tree roots and are known for their peculiar aromas and flavors, which are highly prized in *haute cuisine* [4]. The genus *Tuber* comprises around 200 species [5], the most valuable being *Tuber melanosporum* Vittad. (Périgord black truffle), *T. magnatum* Picco (Italian white truffle), *T. aestivum* Vittad. (Summer or Burgundy truffle), and *T. borchii* Vittad. (Bianchetto truffle) [6].

Beyond their use as food, studies show that truffles potentially have antioxidant properties [7] as well as antiangiogenic and anti-inflammatory traits [8]. They have also been found to exhibit antitumor activity [9]. Truffles are known for their distinctive aroma, attributed to the presence of volatile organic compounds (VOCs) [1,10,11]. Truffle VOCs consist of 30 to 60 volatile constituents, including alcohols, ketones, aldehydes, and aromatic and sulfur compounds, each contributing to the unique scent [1].

The fruiting bodies of *Tuber* can be cultivated either naturally or semi-artificially. Semi-artificial cultivation involves the controlled mycorrhization of suitable tree seedlings using *Tuber* mycelium or spores, which are then planted in appropriate soils [12]. While it is impossible to obtain fruiting bodies in vitro without the host plant, pure cultures of truffle mycelium, including *T. borchii*, can be grown [7,13,14]. Various methods have been developed for industrial in vitro cultivation using natural truffle samples [14]. However, these mycelia typically exhibit slow growth, often taking several months to obtain sufficient biomass for mycorrhization or other studies. As a result, the number of strains characterized for use as model systems remains limited. In vitro studies have demonstrated the presence of VOCs in axenic mycelium cultures of truffles [15,16,17]. The availability of new strains for genetic and physiological studies in this fungus could facilitate progress in this field of research.

*T. borchii* is a whitish truffle found in Europe, especially in Italy, during winter and early spring. It grows in cold temperate to Mediterranean climates, subalkaline, and less frequently in slightly acidic soils, associated with trees and shrubs such as oak, poplar, strawberry tree, and pine [18,19]. Currently, there are 72 pezizomycete genomes available in the Mycocosm database coordinated by the Joint Genome Institute (JGI), with 10 corresponding to *Tuber*. The main features of the truffle genomes were elucidated after sequencing *T. melanosporum* in 2010 [20]. The other sequenced *Tuber* genomes include *T. aestivum*, *T. borchii*, *T. brumale* Vittad., *T. canaliculatum* Gilkey., *T. gibbosum* Harkn., *T. indicum* Cooke and Massee., *T. magnatum*, *T. melosporum* P.Alvarado, G.Moreno, Manjón & J Díez., and *T. mesentericum* Vittad. [21].

The genome sizes of the sequenced truffles are the largest in the *Pezizomycetes* group, with a mean genome size of 114.13 Mbp compared to the 71.45 Mbp of the entire group. Among the truffles, eight of the ten largest genomes belong to this group. The difference becomes even more apparent when comparing the genomes of *Tuber* species, with a mean genome size of 129.48 Mbp. However, the larger genome size of *Tuber* species does not result from having more gene models (with mean numbers of 11,621.33 and 11,776.50 for *Pezizomycetes* and *Tuber*, respectively). However, it is due to an expanded number of transposons and repetitive sequences in truffles, as described in the analysis of the *T. melanosporum* genome [20].

The reference draft genome of *T. borchii* (Tbo3840) was sequenced in 2018 [22]. The nuclear genome assembly of this species is 97.18 Mbp in length, which is relatively small compared to other *Tuber* and truffle genomes. The genome contains 12,346 predicted genes, which align well with the expected number of genes for truffles and *Tuber*. Interestingly, the *T. borchii* genome does not contain genes coding for the glycosylhydrolases GH6 and GH7, indicating its mycorrhizal nature.

Beyond gene catalogs, it is crucial to understand the expression profiles of different metabolic pathways to understand the life cycle of truffles. The transcriptome profile of the central metabolic pathways can serve as an internal reference for evaluating the activity of secondary metabolite production, especially when comparing different strains becomes challenging due to differences in genetic background, environment, and culture conditions. This approach allows a more accurate assessment of the metabolic activity and regulation of truffles [23].

Most transcriptomic studies primarily focus on identifying differences in gene expression between different conditions. However, in many cases, there is limited information available regarding the overall transcriptomic landscape. Essential questions regarding the distribution of transcriptional effort, the range of transcriptional activity across genes, the identification of highly transcribed genes, and similar inquiries are often overlooked. This bias significantly restricts our understanding of the fundamental biology of the organism being studied. We miss crucial insights into the regulatory mechanisms and functional significance of gene expression patterns by neglecting these aspects. Therefore, it is essential to address these gaps to obtain a more comprehensive understanding of the organism’s biology.

We define transcription effort as the percentage of the total transcription corresponding to specific genes or gene groups. Whereas in most sequenced fungal genomes, the models without an assigned function represent nearly 30% of the predicted genes, their transcription effort could represent more than 50% and vary in different conditions. And this is a general rule for other genes and gene groups [24].

Combined genomic, transcriptomic, proteomic, and metabolomic studies have provided comprehensive schemes of the carbon metabolism in fungi for which a gold-standard genome is available and the transfer of these models to related species [25,26]. In the case of *Tuber*, this type of study was pioneered by Ceccaroli et al. [23], who studied more than 100 genes involved in the central carbohydrate metabolism of *T. melanosporum*. These studies have focused primarily on ascertaining the presence of genes coding for the enzymes of the different metabolic pathways and detecting their up and downregulation in comparison between different growing conditions.

In this study, a new strain of *T. borchii* was isolated, and its mycelium was cultured axenically in vitro under various conditions using different culture media. The primary objectives of this work are to provide a molecularly amenable *T. borchii* strain for future studies and to establish fundamental references for its central and secondary metabolism. The study focused on the transcriptional landscape of the entire genome, aiming to answer questions such as identifying highly expressed genes, examining the expression patterns of the genes involved in central metabolisms within the overall transcriptome landscape, and assessing the expression levels of the genes associated with secondary metabolism gene clusters and the production of VOCs. Additionally, the concept of “transcriptional effort” was introduced, representing the percentage of total transcription dedicated to a specific gene or group of genes. The analysis includes genes associated with metabolic pathways and the synthesis of truffle aromas, using the *T. borchii* 3840 genome as a reference. This research paper aims to provide valuable insights into in vitro and in vivo functional studies of this group of organisms. Investigating genetic and metabolic aspects contributes to a better understanding of *T. borchii* and its unique characteristics.

## 2. Results

### 2.1. Culture Conditions, Growth Rate, and Biomass Production

Several truffles (*T. borchii*, *T. aestivum*, *T. magnatum,* and *T. melanosporum*) were obtained from local markets and collectors in different regions of Spain. The truffle surfaces were washed and aseptically broken to transfer gleba fragments to a Petri dish, following the procedures described in the Section 4. However, we experienced difficulties isolating the mycelium from the fruiting bodies used in this study (Figure 1A). Despite our best efforts, isolating mycelium from *T. aestivum*, *T. magnatum*, and *T. melanosporum* samples was impossible. However, in the case of a *T. borchii* sample obtained from collectors in Castilla y León, Spain, at the end of the winter season, mycelium could be successfully isolated after eight days of culture using woody plant sucrose (WPS) as the isolation medium.

Once the mycelium was successfully isolated, the strain was subjected to molecular classification by sequencing the internal transcribed spacer (ITS) region (Accession Number OQ002403, GenBank NCBI database). The obtained sequences were compared to the NCBI database, and the top ten blast hits revealed high similarity to sequences of various *Tuber* species. The closest match was found with *T. borchii* Tbo3840 [22], indicating that the isolated strain belongs to the *T. borchii* species. For this study, the strain is referred to as *T. borchii* SP1 (Figure 1B).

Various culture media were tested to determine the optimal for promoting the highest growth rate of the *T. borchii* SP1 strain in plate cultures. Figure 1C illustrates the growth of colonies after 35 days of incubation at 22 °C in the dark. The cultures cultivated on solid maltose medium (MPY) exhibited the highest growth among all the tested media. This result suggests that the MPY medium provides optimal conditions for the growth of *T. borchii* SP1.

During incubation in submerged liquid cultures, the growth of the *T. borchii* mycelium was only observed in static mode, while agitation at 200 rpm did not support growth. In all cases, the mycelium grew as a large submerged pellet, and the formation of floating biofilms was never observed. Three different culture media were utilized to assess biomass production in liquid cultures of *T. borchii*: woody plant + glucose (WPG), maltose + peptone + yeast extract (MPY), and potato dextrose broth (PDB). Cultures were maintained for 40 days before harvesting the mycelium. Interestingly, the mycelium produced in WPG appeared whiter than in MPY or PDB. Additionally, when the mycelium grown in MPY or PDB was filtered to remove the broth, it quickly darkened (Figure 2).

The mycelial biomass produced after 40 days of submerged static culture is indicated in Table 1. As in the case of solid cultures, the growth in MPY was higher than in the other media.

### 2.2. Hyphal Morphology

The growth and morphological characteristics of *T. borchii* SP1 mycelium cultivated on different solid media (MPY, PDB, and WPG) were analyzed. In all cultures, the hyphae exhibited hyaline coloration, branching, septation, and hyphal anastomosis (Figure 3). The coarseness of hyphae and branching angles were similar among the three media. The average hyphal thickness ranged between 2.7 and 3.7 µm, and the average branching angle varied between 47 and 57 degrees. Regarding cell size, the distance between septa was larger in the mycelium grown on MPY (43.38 ± 16.26) than in PDB or WPG.

### 2.3. RNA-Seq and Transcriptome Analysis

The transcriptomes of four different *T. borchii* SP1 submerged cultures were examined. The results of these analyses are summarized in Table 2. Detailed transcriptome data can be found in Appendix A. Raw data were deposited in the Gene Expression Omnibus (GEO) NCBI database under the accession number GSE233283. The transcriptome reads were aligned to the reference genome sequence of Tubbor1 Tbo3840, available in the Mycocosm database accessible at the Joint Genome Institute website (JGI, https://mycocosm.jgi.doe.gov/mycocosm/home (accessed on 15 April 2022)). Gene expression was detected for over 10^4^ genes in each sample, representing approximately 84.6% (MPY sample) to 88.6% (WPGY sample) of the annotated genes for this species. Functional studies used the Eukaryotic Orthologous Groups of Proteins (KOG) [27] annotation from the *T. borchii* genome site at JGI. The total transcriptome counts were normalized to TPM (total count number normalized to 10^6^ in each sample) to evaluate the transcriptional effort associated with different genes and gene groups.

We define the transcriptional effort as the percentage of the total transcriptome normalized reads associated with a gene or a group of genes. If we order the genes from the most to the less expressed, the cumulative transcriptome effort could be fitted to logarithmic curves with determination coefficients (R^2^) with values of 0.984 for WPG, 0.984 for WPGY, 0.977 for PDB, and 0.979 for MPY. These results indicate that a significant portion of the transcription effort was concentrated on a few genes (see Appendix A.

In the WPG and WPGY samples, three of the 12,346 truffle genes accounted for 10% of the total gene expression. This value was added up by the seven and eight more expressed genes in the PBD and MPY samples, respectively. Furthermore, approximately 75% of the total expression was attributed to around 10% of the annotated genes in the four samples. Notably, less than 2000 genes accumulated to over 80% of the total gene expression in all cases.

The genes that accounted for up to 80% of the total transcription in the four transcriptomes exhibited individual values higher than 100 TPMs, indicating active transcription, albeit at a relatively low level compared to the most highly transcribed genes. The most transcribed genes had normalized expression levels higher than 59,000 in the WPG and WPGY samples and approximately 20,000 in the PDB and MPY samples.

Although nearly 80% of the genes that exhibited transcription contributed individually to a small extent in the overall transcription effort, their cumulative transcription accounted for 20% of the total. In summary, 20% of the genes accounted for 80% of the transcription effort, while the remaining 80% accounted for 20%.

### 2.4. Functional Annotation of the T. borchii SP1 Transcriptome

Approximately 42–44% of the annotated genes in the analyzed samples had a presumed KOG function, whereas the overall percentage of *T. borchii* genes with a KOG annotation was 46.06%. This suggests that the genes responsible for most of the transcriptional effort were more likely to code for unknown functions. Analysis of the top 10 most expressed genes in each sample revealed that most of them had no assigned function, except for genes encoding histones (found in the four samples), chaperones (found in the samples cultivated in minimal media WPG and WPGY), and GAPDH (found in the samples cultivated in complex media MPY and PDB) (Table 3).

The frequency of KOG categories in the expressed genes was investigated, and it was found that the four samples exhibited the same profile (Figure 4A). This similarity was expected since most *T. borchii* genes were expressed in all four samples (See Appendix A).

However, when considering the transcript counts associated with each category (Figure 4B), it was observed that higher transcription efforts were observed in categories J (translation), O (post-translational modification, protein turnover, chaperones), U (intracellular trafficking and secretion), and S (unknown function). The effort associated with category O was higher in the woody-plant-based culture media (WPG and WPGY) compared to the cultures made in complex media (MPY and PDB). On the other hand, categories G (carbohydrate transport and metabolism) and Q (secondary metabolites) showed a higher expression in the MPY and PDB cultures.

### 2.5. Pathways of Central Metabolism

A general description of the characteristics of *T. borchii* SP1 transcriptome was undertaken to serve as a reference for future studies. Gene expression in the central metabolic pathways was assessed to advance further in this direction. Specifically, attention was directed toward four pathways: glycolysis, tricarboxylic acid cycle (TCA), glyoxylate cycle, and electron transport chain (ETC). The transcription of the genes coding for enzymes participating in these pathways was monitored, and the key findings are presented below. This study aimed to determine the transcription effort associated with each of these pathways and whether all the enzymes participating in a pathway were similarly transcribed, as well as compare their transcription levels with enzymes from other pathways.

#### 2.5.1. Central Metabolism Glycolysis Genes

A total of 43 gene models were annotated in the glycolysis pathway of *T. borchii*. These genes encoded 11 enzymes involved in the central pathway and genes coding for alcohol dehydrogenases. Several isozymes were transcribed within these genes, including three hexokinases, two glyceraldehyde-3P-dehydrogenases, nine phosphoglycerate mutases, two pyruvate kinases, and seven alcohol dehydrogenases. However, the expression levels of these genes varied.

In the samples cultivated in complex media (MPY and PDB), all the genes except those coding for glyceraldehyde-3P-dehydrogenase (GA3PDH) and alcohol dehydrogenase (ADH) had expression values equal to or lower than 500 TPMs (Figure 5). These expression values ranked these genes after the top 275–300 most expressed genes in the samples. The less expressed gene coded for phosphofructokinase (PFK, ID 1092436). On the other hand, the gene model coding for the GA3PDH NAD binding domain (ID 957843) ranked as the sixth most expressed gene in the PDB sample and the most expressed gene in the MPY sample, whereas the gene model coding for the catalytic domain of this enzyme (ID 1079835) was the most expressed glycolytic gene in all four conditions. Additionally, the ADH gene (ID 1133725) showed a high expression in the four samples, with expression levels above 1500 TPMs, placing it among the top 75 most expressed genes in each sample. This result suggests that fermentation plays an important role in the biology of the truffle as cultivated in these experiments.

The transcription of glycolysis genes showed a complete correlation (1.000) between the WPG and WPGY media, indicating a strong similarity in their expression patterns. The correlation between the PDB and MPY media was also high (0.911), suggesting a relatively consistent transcription profile between these two conditions. However, it is important to note that the most distinct transcription profile for the glycolysis genes was observed in the MPY medium, indicating notable differences in the gene expression compared to other media. For further details, please refer to the Appendix A.

Finally, it is interesting that the gene coding for phosphoglycerate kinase (PGK, ID 1077976) exhibited a higher expression in the samples incubated in complex media, particularly in MPY, than those cultivated in woody-plant-based media.

#### 2.5.2. TCA and Glyoxylate Cycle

A total of 25 genes were annotated as coding for enzymes involved in the TCA (tricarboxylic acid cycle), encompassing all the enzymes of the cycle. Except for one gene, all exhibited expression levels below 500 TPMs within the range observed for the glycolytic pathway. The gene encoding citrate synthase showed higher expression levels, reaching 600 TPMs in PDB and MPY cultures (Figure 6). The correlation between the expression levels of these genes in different culture media followed a similar pattern to that observed for the glycolysis genes (see Appendix A). 

In summary, the genes involved in the TCA pathways exhibit similar expression levels to those of the central glycolytic pathway. They also rank similarly, around 275–300, among the genes regarding expression levels.

The glyoxylate cycle, which serves as a shortcut to the citrate cycle, involves two enzymes: isocitrate lyase (IL) and malate synthase (MS). In this study, there were two IL genes, and their accumulated transcription levels were similar to those of the genes in the citrate cycle, ranging from 300 to 600 TPMs. However, there was a notable difference in the transcription of the MS gene. In the woody-plant-based media (WPG and WPGY), the MS gene had a transcription level of 36 TPMs, while in the samples grown in rich media (PDB and MPY), it reached 400–500 TPMs.

The transcription correlation between the minimal media samples (WPG and WPGY) was close to 1.0, indicating a strong similarity in gene expression. However, the correlation between the minimal and complex media (PDB and MPY) dropped to 0.5–0.6, suggesting a divergence in transcription patterns between these two media types.

#### 2.5.3. Electron Transport Chain

In the oxidative phosphorylation pathway, 11 genes were annotated as coding for the enzymes involved in the electron transport chain (ETC). These included two genes for NADH–dehydrogenase, two genes for succinate dehydrogenase (which also participates in the TCA cycle), one gene for ubiquinol–cytochrome c reductase, five genes involved in ATP synthesis and H^+^ transport coupling, and one gene coding for an alternative oxidase.

The expression levels of all these genes were similar across the four conditions analyzed. With two exceptions, their expression values ranged from 100 to 700 TPMs, which was consistent with the expression levels of the TCA and glycolytic pathway genes. The two exceptions were the genes coding for the NADH–dehydrogenase complex enzymes, which had expression values lower than 100 TPMs in all four conditions. Please refer to the Appendix A.

In summary, most of the genes involved in the central glycolytic pathway of *T. borchii* exhibit similar expression levels, ranging from 100 to 500 TPMs, and rank among the 275–300 most expressed genes. However, two specific observations highlight distinct metabolic behaviors.

First, the ADH genes were highly expressed in all four conditions, indicating the significance of alcoholic fermentation as a primary metabolic choice for *T. borchii* under the tested conditions. This high expression suggests a preference for alcohol metabolism in this microorganism.

Second, the genes encoding NADH–dehydrogenase showed low expression levels in all four samples. This observation further supports the notion that *T. borchii* relies more on alcoholic fermentation than oxidative phosphorylation for NAD regeneration.

### 2.6. Genes Involved in Secondary Metabolism Pathways—Clusters

Having established the transcriptional landscape of central metabolic genes as a reference, we focused on the relative expression of genes associated with secondary metabolite clusters in *T. borchii* SP1 mycelium samples. Truffles exhibit diverse biological activities and produce secondary metabolites of interest. Specifically, we investigated the expression of non-ribosomal peptide synthetases (NRPS and NRPS-like) and polyketide synthases (PKS and PKS-like) gene clusters.

All the genes identified within these secondary metabolite clusters showed minimal to null expression levels (below 60 TPMs), ranking in positions 2300–2800 of the transcriptomes (see Appendix A). Collectively, they represented only 0.05% of the total transcriptional effort. This indicates that the transcription of these genes was lower compared to the genes involved in central metabolism.

### 2.7. Genes Related to the Synthesis of Volatile Compounds

Finally, we searched for genes associated with the aroma of truffles based on the studies by Martin et al. [20], who identified 92 aroma-related genes in the *T. melanosporum* species. Interestingly, 85 of these genes were expressed in all the samples of *T. borchii* SP1 (see Appendix A). The expression levels of these genes varied widely, ranging from very low (below 10 TPMs) to very high (over 1000 TPMs). Notably, one particular gene (model ID 133725, mentioned earlier about the expression of ADH in central metabolism) encoding a class V alcohol dehydrogenase exhibited the highest expression among all the enzymes involved in aroma production.

Figure 7 illustrates the expression levels of the top ten most highly expressed aroma genes in the four samples, ranging from 200 to 1900 TPMs. These genes encompass a variety of functions and are associated with different metabolic pathways, including aldehyde and alcohol synthesis, sulfur compound metabolism, and isoprenoid synthesis. Notably, the enrichment of genes encoding enzymes involved in aldehyde and alcohol synthesis pathways was observed within this gene group.

## 3. Discussion

### 3.1. SP1 Is A New Mycelial Isolate of Tuber Borchii

Truffles are highly esteemed edible fungi, garnering significant attention in the research field. In recent years, the submerged fermentation of fungal mycelia has emerged as a promising approach to address the laborious and time-consuming nature of truffle cultivation [13,14]. Numerous research teams have studied optimizing the submerged fermentation of truffle mycelia [28,29,30]. These investigations typically assess variations in biomass, exopolysaccharides, enzyme production, metabolite presence, and volatile organic acids, aiming to achieve conditions that closely mimic the natural environment conducive to fruiting body formation.

Nevertheless, the availability of *Tuber* strains for laboratory studies remains limited. Cultivating *Tuber* spp. mycelia in a culture medium is challenging, often resulting in extremely slow growth and minimal biomass yield [31]. Consequently, there is a pressing need to expand the repertoire of *Tuber* strains that can be successfully cultured in vitro and molecularly characterized. This broader range of strains would significantly enhance the scope of research in this field.

This paper presents the isolation and characterization of a novel *T*. *borchii* strain named SP1 obtained from the soil in Castilla-León, Spain. During the isolation process, the truffle was carefully cleaned to eliminate any potential bacterial or fungal contaminants, and samples were taken from the gleba. The original samples were heavily contaminated with various microorganisms, including fungi from the genera *Cladosporium* and *Fomitopsis*, identified as the most prevalent contaminants through ITS sequencing (detailed data not presented). This is common, as ectomycorrhizal truffles and their fruiting bodies often harbor a diverse microbial community of filamentous fungi, yeasts, and bacteria [32,33,34].

The mycelium obtained from this strain was derived from the gleba tissue, indicating that it represents the maternal mycelium of this truffle (specific data not provided) [35,36]. The identification of this newly isolated *T. borchii* strain, SP1, was confirmed by sequencing the ITS region of the isolated mycelium.

In order to overcome the challenge of a low growth rate, various solid culture media, including PDA, MPY, WPGY, and WPG (detailed composition provided in Section 4), were tested for the isolation and cultivation of *T*. *borchii* SP1 mycelium. Among these media, MPY and WPG supported the highest growth rates. Notably, the estimated mycelial growth rate of *T*. *borchii* SP1 in MPY was approximately 40 cm/year, which represents a significant improvement compared to the growth rate reported by Iotti et al. in their study on *T. borchii* [31] (Figure 1C). Additionally, it was observed that the mycelia exhibited better growth when a lower agar concentration was used (specific data not provided).

On the MPY culture medium, the mycelium of *T. borchii* SP1 exhibited white colonies composed of straight, septated, and sparsely distributed hyaline hyphae (Figure 1B). Occasional branching and hyphal anastomoses were observed (Figure 3). The morphological characteristics of the mycelium cultivated in the three different culture media were similar. However, it was noted that the mycelium grown on maltose-containing medium had larger cells, indicating that the composition of the culture medium can influence hyphal morphogenesis [37]. Notably, no vesicles, which are typically observed in *Tuber* spp. mycelia [31,38] were observed in *T. borchii* SP1.

To identify the optimal conditions for achieving a higher biomass yield, submerged cultures of *T. borchii* SP1 were conducted using three different media, WPG, MPY, and PDB. It is worth noting that all cultures were maintained in static conditions, as no growth was observed under shaking conditions at 200 rpm. This observation is consistent with a previous study by Lacourt et al. in 2002, which also conducted submerged cultures of *T*. *borchii* under static conditions [7], but differs from the results reported by Chen et al. [39] who obtained growth under shaking conditions. The lack of growth under shaking conditions that we observed could result from mechanical disturbances on the mycelium or a preference for microaerophilic conditions, which could be consistent with a fermentative metabolism, as discussed below.

The culture medium that resulted in a higher biomass yield for *T. borchii* SP1 was MPY. When examining the hyphae under the microscope, it was observed that the cell sizes were larger compared to the other culture conditions. Additionally, the transcription of certain key genes involved in the central metabolism was also found to have higher values in the MPY culture medium compared to other conditions. This suggests that MPY provides a favorable environment for the growth and metabolic activity of *T*. *borchii* SP1.

The findings of Amicucci et al. (2010) [40] regarding the influence of carbon sources on *T*. *borchii* mycelium growth, specifically observing better growth in glucose compared to maltose or sucrose, differ from the results obtained with *T. borchii* SP1 in this study. In the case of *T. borchii* SP1, the mycelium exhibited better growth in the presence of maltose rather than glucose. These contrasting observations suggest that different *T. borchii* strains may respond differently to various carbon sources, highlighting the potential variability within the species.

The growth of *T. borchii* mycelium as a single pellet colony that occupies the culture flask is a common observation in submerged cultures. The merging of pellets into one can occur due to the aggregation of individual mycelial masses during growth. The color of the pellet can vary depending on the culture medium used, indicating potential differences in metabolic activity or pigment production. Figure 2 visually represents the pellet morphology and color variations in different culture media.

In some cases, when the mycelia were collected by filtration, they turned brownish rapidly. This color change was more evident when it occurred in the cultures performed in MPY or PDB than in the cultures made using WPG. The brownish of the mycelium in these samples could be correlated with the slightly higher expression of the genes in the clusters of secondary metabolism in the cultures made in complex media (see below).

In summary, a new strain of *T. borchii* that can be axenically cultured in maltose-containing plates and static-submerged cultures was isolated. This strain can be maintained by subculturing indefinitely without signs of strain degeneration.

### 3.2. RNA-Seq and Transcriptome Analysis

The transcriptome analyses of the *T. borchii* SP1 strain described in this paper are preliminary and qualitative and do not pretend to be an exhaustive quantitative study since the difficulties of obtaining enough RNA material prevented making replicas of the different conditions studied. Our objective was to study the expression of different genes and gene families under different culture conditions and to identify common expression patterns that can be used as a reference for future comparative studies. In this context, we assume that all the mRNA species within a given sample are extracted and processed equally, and the comparative expression values in a given sample provide information that can be contrasted with other similar from other different samples. Consequently, transcription profiles observed in different experiments could represent consistent transcriptional effort distributions and inform about primary metabolic processes in the organism.

The transcriptomic profile of *T. borchii* SP1 growing in static liquid cultures of different media (PDB, MPY, WPGY, and WPG) was studied. The samples were harvested after 64 (WPGY and WPG) or 85 days of culture (PDB and MPY). The expression of 10.4 × 10^3^ to 10.9 × 10^3^ gene models was detected and identified using the published *T. borchii* Tbo3840 genome as a reference in the four media. These values represent between 84.6 and 88.6% of the annotated genes.

The most expressed gene in the WPG and WPGY samples was ID 990338, with an expression level representing 5.97% of the transcription effort in these conditions. Surprisingly, the transcription level of this gene in the PDB and MPY samples was much lower and represented only 0.19 and 0.28% of the transcription effort, respectively. Gene ID 990338 codes presumably for an Hsp26/Hsp42 chaperone. This chaperone belongs to the family of the Hsp20 and is a small heat-shock protein that suppresses protein aggregation and protects against cell stress. Hsp20 has been reported to be absent in the *T. melanosporum* genome and other ascomycetes [41]. However, the search for BlastP hits of this protein reveals its presence in *T*. *brumale*, *T. aestivum*, *T*. *indicum*, *T. magnatum*, and many other ascomycetes. This protein was used to construct a temperature-tolerant strain of *Lentinula edodes* [42] and is found in tandem repeats in a temperature-adapted *Coriolopsis trogii*. The model identified in *T*. *borchii* corresponds to a secreted protein. Secreted heat shock proteins are proposed to act as intercellular signals [43]. Their function in *Tuber* is unknown.

The most expressed gene in the PDB samples codes for a protein without an assigned function (ID 1067414). The transcription effort of this gene in PBD is 2.10% and 1.37% in MPY. The gene coding for this protein is also among the most expressed in WPG and WPGY, representing 0.96 and 0.5% of the gene expression and among the 10 most expressed genes in these samples. This gene codes for a small (154 amino acids) presumably secreted protein. Proteins similar to this have been found in other ascomycetes (preferentially other *Tuber*) and some budding yeasts. It has a peculiar primary structure as it contains 14 Ser and 34 Thr residues.

Finally, the most expressed gene in MPY was ID 957843, coding for the glyceraldehyde 3-phosphate dehydrogenase (GA3PDH) NAD binding domain. The portion of the total transcription associated with this gene was 1.95, 1.36, 0.35, and 0.35% in the MPY, PDB, WPG, and WPGY samples, respectively. In the two samples derived from complex media (MPY and PDB), this gene was among the ten most expressed, whereas in the samples derived from minimal media (WPG and WPGY), the gene was among the 30 most expressed.

### 3.3. Comparison of the Transcriptomes from the Different Culture Media

The correlation between the normalized expression values for all genes in all the pairwise combinations of conditions (Appendix A) was studied. The correlation between the gene expression values when WPG and WPGY samples were compared was 99.6%, with R^2^ being 0.996, thus indicating that both samples were virtually identical. No genes in this comparison fell outside the twofold limit established as a rough comparative criterion. This result suggests that the yeast extract in the WPGY medium did not induce significant gene expression changes in the samples. When all other comparisons were made, the R^2^ values were lower (between 0,64 and 0,70), with the sample from the MPY culture appearing to be the most different. Only a few genes fell out of the twofold expression limits established for these comparisons, most without functional KOG annotation.

The WPG and WPGY samples showed a more similar transcriptome profile than the MPY and PDB samples. There are two possible explanations for this. The woody-plant-based culture media (WPG and WPGY) are synthetic culture media with glucose as the sole carbon source, whereas the PDB and MPY media are undefined culture media with complex carbon sources (maltose and potato dextrose). Furthermore, the WPG and WPGY samples were harvested after 65 days of culture, whereas the MPY and PDB samples were harvested on day 85. Despite these long cultivation times, the high expression of histone-coding genes in the four samples (see Table 3) suggests that the mycelia were actively growing in the four cultures, although at a very low rate.

The transcription efforts of the genes annotated within the O KOG category (post-translational modification, protein turnover, and chaperones) in WPG and WPGY media were 15.27 and 15.30%, respectively, whereas, in the PDB and MPY media, they were 6.43 and 7.61%, respectively. The importance of the transcription of gene ID 990338 in WPG and WPGY has been described above. This gene codes for a chaperone belonging to KOG Class O, and its contribution to the effort of this class is around 6%. Putting aside the contribution of this gene to the complete transcription of class O, this class is still more transcribed in the minimal media (an effort close to 10%) than in the more complex media. Within the 20 most expressed genes in WPG and WPGY, three code for heat shock proteins (Hsp20 and Hsp70) and one for a ubiquitin-like protein.

Concerning the genes belonging to the KOG class G (carbohydrate transport and metabolism), the observed enrichment in the expression of the genes involved in carbon source processing could be a consequence of the complexity of these two media compared with the glucose used as the sole carbon source in WP-based broths.

Finally, the expression of the genes classified in the KOG class Q (secondary metabolites biosynthesis, transport, and catabolism) seems higher in the MPY and PDB samples than in the WP-based ones. This result suggests that the mycelium cultivated in nutritionally richer broths can produce more secondary metabolites than in the basal WP medium. This result is essentially a consequence of the expression of the ADH genes.

### 3.4. Pathways of Central Metabolism

The NADH–dehydrogenase system is critical for regenerating the NAD^+^ required for the glycolytic reactions. The combination of the low expression of the NADH–dehydrogenase system combined with the high expression levels of the alcohol dehydrogenase in the four media (see Figure 5) suggests that this fungus displays a metabolism preferentially fermentative in these aerobic conditions. In yeasts, alcoholic fermentation is known as the Crabtree effect in aerobic conditions. This effect occurs by inhibiting the TCA and electron transport chain in conditions of a high glucose concentration. The amount of glucose used in our system could be high enough to trigger this effect in *T. borchii*, preventing the accumulation of biomass in this way. Chen et al. [39] observed the best growth in sucrose over glucose in submerged cultures of *T. borchii*.

Ceccaroli et al. [23] reporteded the presence of an acid invertase gene in *T. melanosporum*. This enzyme breaks down sucrose and could explain the performance of this fungus using that sugar as a carbon source. *T*. *borchii* SP1 genome contains one invertase gene that is expressed at a very low level (lower than 10 TPMs) in all the samples analyzed. In addition to the invertase, *T. borchii* codes for three trehalase genes that are also expressed at low levels (below 50 TPMs) except one of them (ID 1130405) more expressed in the samples cultivated in minimal medium.

Finally, the glyoxylate cycle has been extensively studied in *T*. *borchii* [44]. In the paper by Ceccaroli quoted above, the authors report the upregulation of the expression of the malate synthase gene in the root tips compared to the mycelium. Our results indicate that the basal level of expression of this gene in our culture conditions is higher in the cultures performed in complex media in comparison to those made in woody-plant-based broths.

Besides the high level of ADH expression, the expression of GAPDH, PGK, and PK (responsible for phosphorylation at the substrate level) was also high, especially in the MPY culture medium. The high quantities of PGK and PK produce ATP in more significant amounts, which could justify the better mycelial performance in the culture MPY medium.

### 3.5. Genes Involved in Secondary Metabolism—Clusters

Secondary metabolites are not produced in the rapid growth phase (trophophase) but rather during the later production phase (idiophase). Generally, the synthesis of these metabolites begins when a nutrient source is depleted, such as carbon or nitrogen [45]. Although the transcriptomic analysis of this study was carried out with samples from days 65 and 85 of growth, it is possible that these periods were insufficient to produce and accumulate secondary metabolites. As it has been discussed above, the high expression of genes coding for histones in the four samples supports this assumption.

While the genes coding for enzymes involved in the central metabolic pathways are dispersed across the genome, those coding for enzymes participating in the secondary metabolism are arranged in biosynthetic clusters [45]. Ascomycete genomes encode an average of ten non-ribosomal protein synthetases (NRPS), sixteen polyketide synthetases (PKS), two tryptophan synthetases (TS), and two dimethylallyl tryptophan synthases (DMATS) per genome [46]. The automatic annotation of the *T*. *borchii* genome used as a reference for this study, however, identifies seven clusters coding for highly conserved proteins: one non-ribosomal peptide synthetase (NRPS, containing proteins IDs 1077522, 1099828, and 1122331), one type I polyketide synthase (PKS, ID 962914), three NRPS-like (containing proteins IDs 1126844, 966209, 1119995, and 1032817) some of them with PKS domains, and two PKS-like (containing protein ID 1116892 and proteins 1121687, 970716, 1076327, and 970641). This number and distribution are also found in the genome of *T. melanosporum.* This reduced number of secondary metabolite clusters in the *Tuber* genomes could result from a miss-annotation of some of them in the highly complex genomes of these species.

The expression of the genes of the secondary metabolism clusters was low (lower than 50 TPMs in all the samples), except for two PKS genes in the PDB cultures that showed around 150 TPMs. Interestingly, the mycelium harvested under these conditions showed a more intense brownish color that could correlate with the production of compounds that are easily oxidized upon removing the liquid culture medium. Polyketides are present in the black allomelanin pigments formed by truffles [47].

### 3.6. Genes Related to the Synthesis of Volatile Compounds

Regarding genes related to truffle aroma, it was observed that most of the analyzed genes were present and expressed in the samples. Upon analysis of the ten genes with the most significant expression, it was found that the gene expression was very similar in WPG and WPGY media, suggesting that adding yeast extract did not seem to influence the expression of these genes. It was verified that the PDB sample in some genes presented a higher gene expression when compared to the other media. It was also observed that genes were expressed very differently between the different growing conditions. These results explain how the substrate’s composition can influence the production of VOCs by valuable *Tuber* spp. species [29]. PDB, the culture broth that supported the lower biomass yield, showed the highest expression of genes related to aroma production. It is most likely that the expression of these genes increases in non-optimal cultural conditions, as well as the genes in the cluster of secondary metabolism. It was shown that abiotic interacting stress factors influence the relative expression of genes related to mycotoxins production, a category of secondary metabolites produced by some pathogenic fungi [48].

The gene corresponding to phosphoadenosine phosphosulfate reductase (thioredoxin reductase) was expressed almost five times more in PDB compared to the MPY medium. The thioredoxin reductase was included in this group because it appears in the list of aroma-related genes published by Martin et al. [20]. Thioredoxin reductase also plays a role in protecting the cell against the oxidative stress produced by ROS. As discussed above, the high expression of ADH and the low of the NADH–dehydrogenase genes suggest a fermentative metabolism for *T. borchii* under the culture conditions used. Fermentative metabolism and hypoxia conditions generate high levels of ROS. The data presented here suggest that the production of thioredoxinreductase could help prevent the detrimental effects of these toxic compounds.

## 4. Materials and Methods

### 4.1. Strain and Culture Conditions

Several truffles (*T*. *borchii*, *T. aestivum*, *T*. *magnatum*, *T*. *melanosporum*) were obtained from local markets and collectors in different regions of Spain. The truffle surfaces were washed with soapy water (1:200 dilution), brushed to remove excess soil, and dried with absorbent filter paper. Then, the truffles were aseptically broken in two halves, and small pieces of the innermost part of the gleba were transferred to Petri dishes with different culture media supplemented with amoxicillin at 100 mg/mL to avoid bacterial contamination. All cultures were incubated at 22 ± 1 °C in the dark.

The cultivation process was performed in the following culture media: (i) potato dextrose broth 20 g/L (PDB) (Scharlab) or potato dextrose agar (PDA). (ii) Woody plant (WP) medium (composition per liter: 0.2 g KH_2_PO_4_, 0.1 g CaCl_2_·2H_2_O, 0.3 g MgSO_4_·7H_2_O, 0.9 g K_2_SO_4_, 0.1 g myoinositol, 2.3 g woody plant, 1 mL oligo-elements solution containing, per liter, 22.3 mg MnSO_4_·H_2_O, 0.014 mg FeSO_4_·7H_2_O, 8.6 mg ZnSO_4_·7H_2_O, 0.25 mg Na_2_MoO_4_·2H_2_O, 0.025 mg CuSO_4_·5H_2_O) [28] was modified. The woody plant was occasionally supplemented with 20 g/L glucose (WPG), 20 g/L glucose plus 5 g/L yeast extract (WPGY), 20 g/L dextrose (WPD), 20 g/L dextrose plus 5 g/L yeast extract (WPDY) or 20 g/L sucrose (WPS). (iii) Maltose medium (MPY) containing, per liter, 35 g maltose, 5 g peptone, 5 g yeast extract, 0.5 g MgSO_4_·7H_2_O, 1 g K_2_SO_4_, and 0.05 g myoinositol [43] was modified. The pH of all the culture media was adjusted to 6.3–6.4 with NaOH. All liquid cultures were incubated in the dark under static conditions at 22 ± 1 °C.

When needed, 7 g/L agar was added for solid cultures. Maltose and yeast extract were purchased from Scharlab (Barcelona, Spain), peptone from Panreac (Barcelona, Spain), woody plant from Merck (Madrid, Spain), and agar from Scharlab.

### 4.2. Growth Rate

The linear growth rate was measured as follows: one plug (0.2 cm^2^) of actively growing mycelium was placed in the center of the Petri dish, and the linear growth of the colony’s edge was measured in the four perpendicular directions each week until the 35th day of growth. All plates were made in triplicate to calculate the mean growth rate values and standard deviation. An ANOVA and Tukey’s test were used to compare variations between the means of the different groups.

### 4.3. Biomass

Three plugs (0.2 cm^2^) of actively growing mycelium from the outermost growth zone of a solid culture were used to inoculate 250 mL flasks containing 100 mL of medium. The flasks were grown at 22 °C in the dark and under static conditions for 40 days. After 40 days, the mycelia were filtered through a micropore filter, weighed, and dried at 50 °C for 24 h, when they were weighed again to obtain their fresh and dry weight. The weight of the agar plugs was discounted. There were four replicates for each culture medium.

### 4.4. Molecular Characterization

The mycelium was ground using liquid nitrogen. DNA extraction was performed using the commercial EZNA Fungal D.N.A. Mini Kit (Promega Biotech, Norcross, GA, USA) following the producer’s protocol. After checking the DNA concentration, a PCR was performed using sequences from the internal transcribed spacer (ITS) region, primers ITS4 5′-TCCTCCCGCTTATTGATA-3′ and ITS1F 5′-CTTGGTCATTTAGAGGAAGTAA-3′1. The PCR conditions were as follows: An initial denaturation step at 95 °C for 5 min, followed by 29 cycles consisting of 1 min denaturation at 95 °C, 0.5 min annealing at 53 °C, and 1 min extension at 72 °C. After the last cycle, a 10 min final extension step at 72 °C was added to complete the reaction. The PCR products were sequenced, and the sequences were blasted against the NCBI database to identify the most similar entries. The ITS sequence was deposited in the GenBank NCBI database with accession number OQ002403.

### 4.5. Morphology

Morphological data of the mycelium of *T. borchii* strain SP1 were studied under light microscopy in different culture media. Small pieces (1 cm^2^) of mycelium were analyzed and collected from each Petri dish (three per substrate) after 40 days of mycelial growth. The following parameters were measured: thickness, branching angle, and distance between the septa of the hyphae grown in the different culture media. Photos were taken using a camera (ZEISS Axiocam 208 color/202) attached to the microscope and were further processed using the program ZEN 2.3—blue edition.

### 4.6. RNA seq

For the transcriptome analysis of the SP1 strain, axenic cultures of the mycelium were performed under different culture conditions. The mycelia were collected after 65 days in WPG and WPGY or 85 days in PDB and MPY of the culture.

The mycelial samples were ground using liquid nitrogen and stored at −80 °C for transcriptome analysis. A fungal RNA EZNA kit (Omega Bio-Tek, Norcross, GA, USA) was used to extract the total RNA. R.N.A. quality was determined using electrophoresis on 1% (*w*/*v*) agarose gels. A Qubit^®^ R.N.A. Assay kit (Invitrogen, Life Technologies Corporation, Norcross, GA, USA) was used to measure the R.N.A. concentration. The mRNA libraries were built using TruSeq^®^ R.N.A. (Illumina, Inc., San Diego, CA, USA) following the manufacturer’s instructions. Sequencing was performed with an Illumina NovaSeq 6000 System (North Caroline genomic lab, Raleigh, NC, USA) using paired-end reads of 150 bp.

### 4.7. Transcriptome Data Analysis

The quality of mRNA-seq data was checked using FastQC [49] and trimmed with Trimmomatic [50] to remove reads and sequences containing adapters. The results of the reads from all libraries were mapped to the *T*. *borchii* Tbo3840 reference genome obtained from the JGI Mycocosm platform (https://mycocosm.jgi.doe.gov/mycocosm/home (accessed on 8 November 2022)) using the STAR Galaxy Version 2.7.8 program [51]. The feature Counts [52] program was used to assign the mapped reads generated from RNA sequencing. KOG (EuKaryotic Orthologous Groups) and EggnogMapper [53] were used for protein function annotation. The Kallisto program was used to normalize the read data, which generated values of TPMs that were transformed into per gene [54].

### 4.8. Analysis of Metabolic Pathways

An analysis of the metabolic pathways, glycolysis, citrate cycle (T.C.A.), glyoxylate cycle, and oxidative phosphorylation was performed based on the KEGG annotation of the *T*. *borchii* genome available in the mycocosm portal of the Joint Genome Institute. (https://mycocosm.jgi.doe.gov/Tubbor1/Tubbor1.home.html (accessed on 13 December 2022)).

## 5. Conclusions

A new Spanish strain of *T. borchii* was isolated, the axenic culture conditions were described, and the gene expression in different medium conditions was analyzed. A significant gene expression was obtained for most of the annotated genes, although most of the transcription effort was concentrated in a few genes (80% of the transcription was due to 20% of the genes). The transcription profiles of the cultures made in minimal medium (WPG and WPGY) were highly similar, whereas there were more differences in the cultures made in complex media (PDB and MPY). Altogether, the analyzed transcriptomes can shed some light on the metabolic processes in *T*. *borchii* under the conditions studied. The distribution of gene numbers and expression by KOG categories depends on the culture medium. The transcription of genes involved in the central metabolic pathways was relatively constant in the four conditions studied, providing a reference for other experiments. The high transcription level of ADH genes suggests a fermentative metabolism for *T. borchii*. The reduced expression level of NADH-dehydrogenase strengthens this hypothesis. The in vitro culture of truffle mycelium may become a potential alternative resource for secondary metabolite production and biotechnological applications and for studying the factors required for enhancing its vegetative growth. It may also be related to the production of medicinally important molecules such as polysaccharides. The expression of the genes related to the synthesis of VOCs showed that the substrate composition is a crucial factor in modulating the aroma production from the mycelium. The in vitro culture of genetically selected fungal strains in optimized substrates may become a potential alternative resource for producing natural truffle flavors to substitute synthetic truffle aromas that consumers do not appreciate.

## Figures and Tables

**Figure 1 ijms-24-10981-f001:**
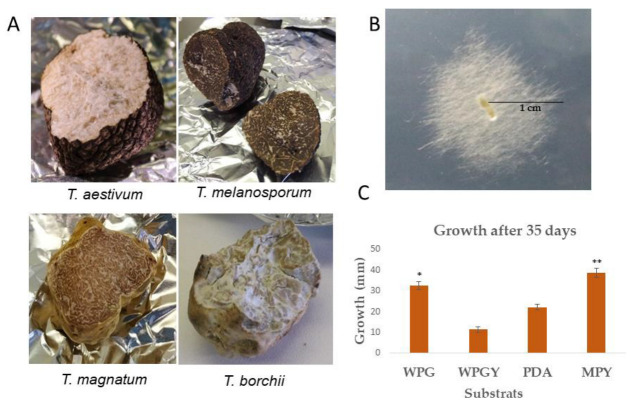
(**A**) Fruiting bodies of the *Tuber* species used in this study. (**B**) Mycelium growth of *T. borchii* SP1 after eight days of culture in woody plant sucrose. The diameter of the colony in the picture is 2 cm. (**C**) Linear growth of *T. borchii* SP1 in different culture media. (* There is a statistically significant difference between maltose+peptone+yeast extract (MPY) and potato dextrose agar (PDA) and woody plant+glucose+yeast extract (WPGY) *p* < 0.05. ** There is no statistically significant difference between MPY and woody plant+glucose (WPG), *p* > 0.05).

**Figure 2 ijms-24-10981-f002:**
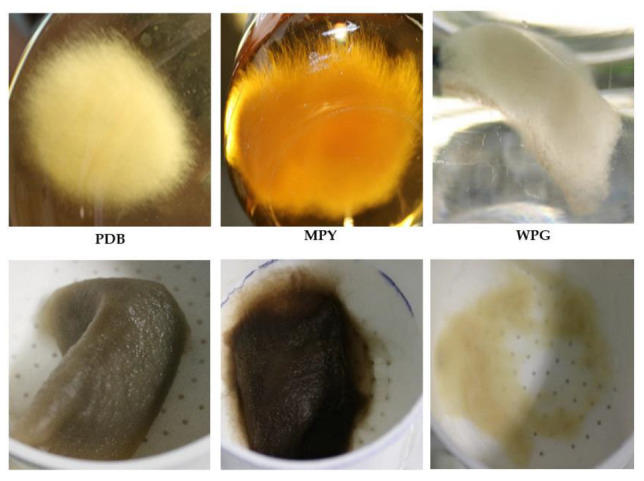
The vegetative mycelium of *T. borchii* SP1 was cultivated in submerged static cultures using potato dextrose broth (PDB), maltose + peptone + yeast extract (MPY), and woody plant + glucose (WPG) media (**upper row**). Subsequently, the same mycelium was filtered to remove the culture broth (**lower row**).

**Figure 3 ijms-24-10981-f003:**
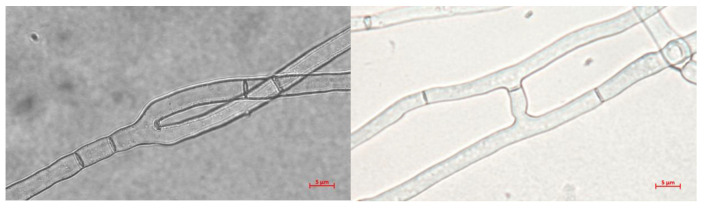
Representative hyphae of *T. borchii* after 40 days of growth (40×). Medium MPY (**left**) and WPG (**right**).

**Figure 4 ijms-24-10981-f004:**
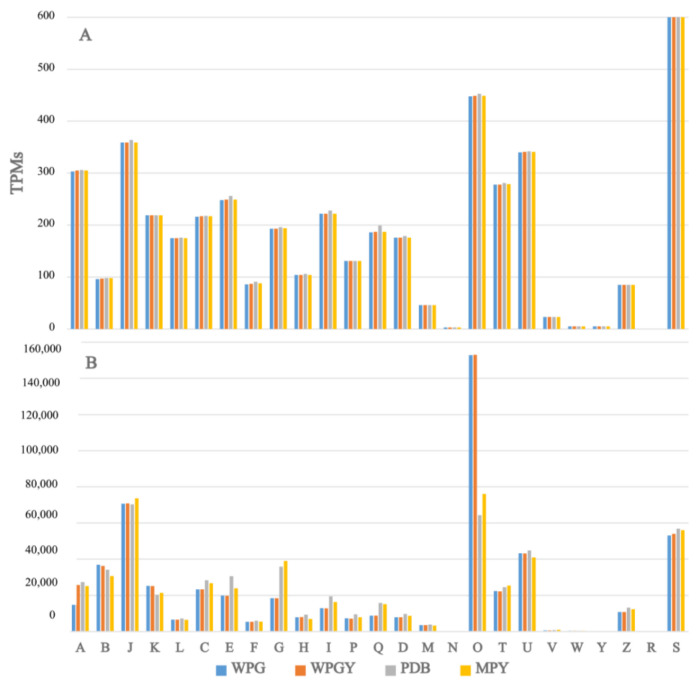
(**A**) The number of genes in each KOG category in the four samples analyzed. (**B**) Number of TPMs reads associated with each one of the KOG categories in the analyzed samples. A—RNA processing and modification, B—chromatin structure and dynamics, J—translation, K—transcription, L—replication and repair, C—energy production and conversion, E—amino acid metabolism and transport, F—nucleotide metabolism and transport, G—carbohydrate metabolism and transport, H—coenzyme metabolism, I—lipid metabolism, P—inorganic ion transport and metabolism, Q—secondary metabolites biosynthesis, transport and catabolism, D—cell cycle control and mitosis, M—cell wall/membrane/envelope biogenesis, N—cell motility, O—post-translational modification, protein turnover, chaperone functions, R—general function prediction only, S—unknown function, T—signal transduction, U—intracellular trafficking and secretion, V—defense mechanisms, W—extracellular structures, Y—nuclear structure, Z—cytoskeleton.

**Figure 5 ijms-24-10981-f005:**
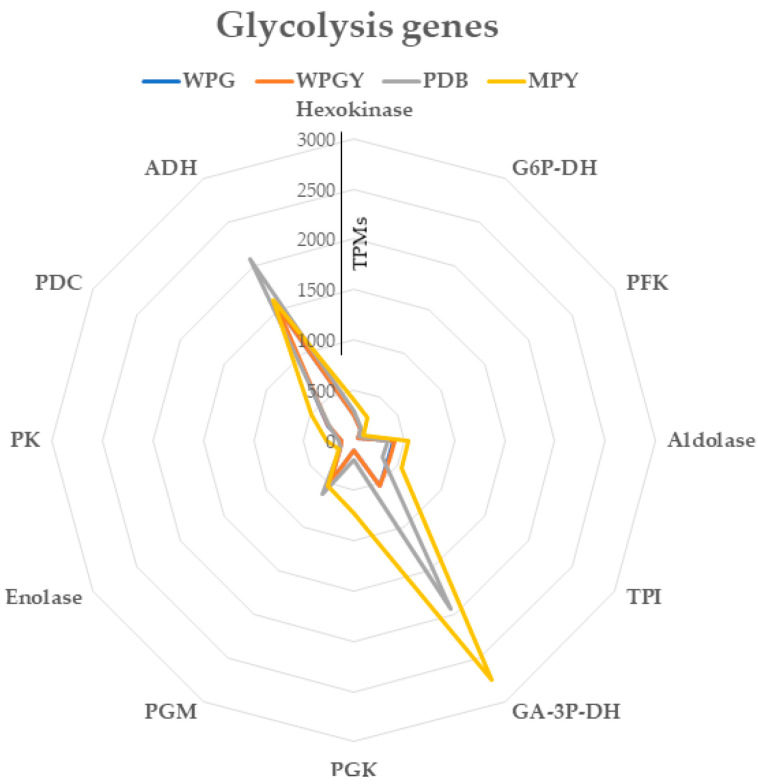
The radar chart represents the transcription of the genes coding for glycolytic enzymes in the four culture media.

**Figure 6 ijms-24-10981-f006:**
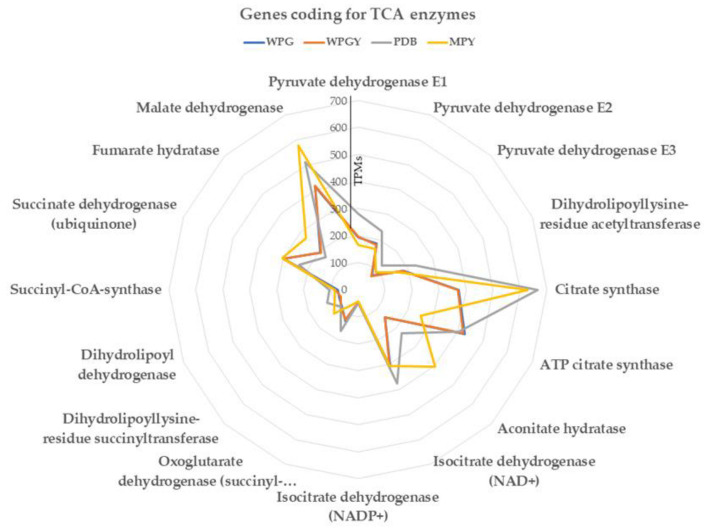
The radar chart represents the transcription of the genes coding for enzymes of the citrate cycle in the four culture media.

**Figure 7 ijms-24-10981-f007:**
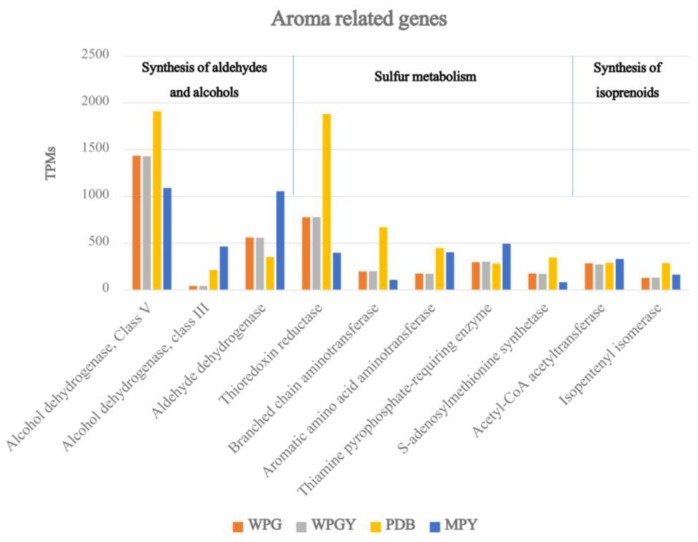
Genetic expression of the ten most expressed genes related to truffle aroma.

**Table 1 ijms-24-10981-t001:** Mean (four repetitions) and standard deviation of the fresh and dry weight of *T. borchii* SP1 mycelium after 40 days of submerged static growth in different liquid culture media.

Culture Medium	Fresh Weight (mg)	Dry Weight (mg)
WPG	535 ± 103	41 ± 7
PDB	470 ± 89	38 ± 1
MPY	1600 ± 270	127 ± 22

**Table 2 ijms-24-10981-t002:** Summary of the RNAseq experiments. The number of expressed genes and genes with Eukaryotic Orthologous Groups of proteins (KOG)—annotation in the four samples.

Sample	Total Reads	Genes Expressed	Genes KOG
WPG	116,376,579	10,870	4645
WPGY	125,457,058	10,941	4642
MPY	47,969,567	10,442	4582
PDB	157,335,190	10,599	4621
Number of genes in the reference genome		12,346	5686

**Table 3 ijms-24-10981-t003:** Functional annotations of the 10 most expressed *T. borchii* genes in the four cultures.

Protein Id	TPMs-WPG	Function	Protein Id	TPMs-WPGY	Function
990338	59710.6	Molecular chaperone	990338	59732.8	Molecular chaperone
1125059	37904.9	N/A	1125059	38565.1	N/A
985714	19175.7	N/A	985714	19689.8	N/A
986569	18513.8	N/A	986569	18720.1	N/A
907400	15291.6	N/A	907400	15059.6	N/A
1138063	12188.2	N/A	1138063	12331.1	N/A
1039918	11559.9	Molecular chaperone	1039918	11506.3	Molecular chaperone
1067414	9526.58	N/A	961630	9712.31	Histone H4
961630	9394.31	Histone H4	1067414	9622.65	N/A
1124376	9124.24	Histones H3 and H4	1124376	9300.14	Histones H3 and H4
**Protein Id**	**TPMs-PDB**	**Function**	**Protein Id**	**TPMs-MPY**	**Function**
1067414	20972.3	N/A	957843	19500.8	GAPDH
1138063	18989.4	N/A	892438	18481.5	N/A
1098806	16159.1	N/A	1067414	13726.7	N/A
1125059	13738.2	N/A	1098806	11293	N/A
957843	13582.2	GAPDH	987363	10269.6	N/A
907400	12134	N/A	1138063	9857.26	N/A
1124376	10231.8	Histones H3 and H4	1041746	9720.97	N/A
1103054	9425.23	N/A	1124376	8576.88	Histones H3 and H4
961630	9060.03	Histone H4	1127611	8393.75	N/A
981017	6404.74	N/A	907400	8022.94	N/A

## Data Availability

The ITS sequence is deposited under Accession Number OQ002403 (GeneBank NCBI database). The raw transcriptome data are deposited under accession number GSE233283 (GEO NCBI database).

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
