# Peer review of "Transcriptome Metabolic Characterization of Tuber borchii SP1—A New Spanish Strain for In Vitro Studies of the Bianchetto Truffle"

_ijms, 2023, doi:10.3390/ijms241310981_

Round 1

Reviewer 1 Report

In this study, the authors isolated a new Spanish strain, named Tuber borchii SP1, and compared its transcriptomes when grown in four different media. While the purpose of the study is suitable for publication, there are several issues with how the transcriptome data is presented. Firstly, all raw data should be deposited in a public database, such as the SRA in NCBI, and accession numbers provided in the manuscript. Secondly, the authors may not have enough knowledge on how to properly analyze and interpret transcriptome data, resulting in the use of inappropriate terminology and awkward phrasing throughout the manuscript. Moreover, the tables and figures related to transcriptome data need improvement, and it is suggested that the authors review other similar works and prepare new tables and figures accordingly. It is imperative that the manuscript undergo rigorous revision before publication, as simply making minor modifications will not suffice.

Minor comments are as follows.

L36: "spp." should be "spp"

L44: "L44 volatile compounds (VOCs)" should be "volatile organic compounds (VOCs)"

L91: "isolate" should be "strain"

L100: "Petri" should be "petri dish"

L105: WPS should be spelled out

L108: ITS should be spelled out and briefly explained as "internal transcribed spacer", and the nucleotide identity of the sequenced ITS region should be provided to the known sequence

Figure 1: Arial with bold font should be used, "P" should be italicized in the legend, and media in Figure 1C should be described with full names and all abbreviations spelled out in the legend

L134: degree sign should be checked

L139: culture media should be described with full names

L139 and L142: "PDB" is correct, not "PBD", and media in Figure 1C should be consistent with the rest of the manuscript

Figure 2: it should be indicated which images are from the upper or bottom portion of the plate

Table 1: the thresholds for selecting the number of expressed genes should be described

Table 1: "KOG" should be spelled out

L182-184: the two sentences should be revised for clarity as they are unclear

Figure 4: it can be deleted and replaced with a table showing 10 representative genes with high expression based on TPM values from each transcriptome

Table 3: it should be deleted

Figure 5: the quality should be improved, and the meaning of each alphabet in the graph should be explained

Figures 6, 7, and 8: the quality should be improved.

This manuscript should be edited by a professional English editing service. 

Author Response

In this study, the authors isolated a new Spanish strain named Tuber borchii SP1 and compared its transcriptomes when grown in four different media. While the purpose of the study is suitable for publication, there are several issues with how the transcriptome data is presented.

Major comments

Firstly, all raw data should be deposited in a public database, such as the SRA in NCBI, and accession numbers provided in the manuscript.

The raw data have been deposited in a public database as required. The accession numbers are indicated in the main text.

Secondly, the authors may not have enough knowledge on how to properly analyze and interpret transcriptome data, resulting in the use of inappropriate terminology and awkward phrasing throughout the manuscript.

This paper deviates somewhat from the classic transcriptomic reports in its purpose and methodology. In this work, our focus is not on comparing transcription profiles among different samples but instead on describing the transcription profile of a specific organism. Comparisons can provide valuable insights when a reference is available, but they may overlook relevant information without such a reference. As exemplified in our paper, let us consider comparing ADH gene transcription and other glycolysis genes. The ADH genes exhibit significantly higher expression than any other glycolytic gene (except GAPDH in specific samples), but their expression remains consistent across all four analyzed samples. This finding would be overlooked in the traditional analysis of Differentially Expressed Genes (DEGs), as they do not exhibit differential expression between samples. However, we consider this information crucial for understanding the metabolism of this fungus under these specific culture conditions. This unique approach also impacts the graphical representation of the data, as our focus is on highlighting internal differences within a sample.

This analysis has a methodological assumption: all mRNAs within a given sample are equally extracted and processed. Biological and technical replicates are primarily employed to account for variations arising from physiological conditions or experimental fluctuations. In our case, we compare transcription profiles and identify genes more highly expressed than others within the same sample. Our objective is not to identify differences between different conditions but to determine whether genes involved in central or secondary metabolite pathways exhibit similar levels of expression and their contribution to the overall expression as a percentage.

We introduce the concept of "transcriptional effort" as a measure of the proportion of total transcription corresponding to the expression of a specific gene or group of genes. While we acknowledge that variations in the expression of specific transcription factors, for example, can have significant transcriptional and metabolic effects, we believe it is essential not to overlook the cellular effort expended on transcribing and potentially translating other non-master genes. Moreover, many of these highly expressed genes consume substantial cellular energy during transcription and remain functionally uncharacterized orphan genes.

Moreover, the tables and figures related to transcriptome data need improvement, and it is suggested that the authors review other similar works and prepare new tables and figures accordingly.

We have reviewed and modified the tables and figures to align them as closely as possible with the formatting and patterns commonly used in other transcriptome papers.

It is imperative that the manuscript undergo rigorous revision before publication, as simply making minor modifications will not suffice.

We have thoroughly revised the paper and made corrections to improve the English usage based on the suggestions from the referee and the considerations mentioned earlier.

Minor comments

As the manuscript has been rewritten, the line numbers have changed.

(New L39)  L36: "spp." should be "spp":

Corrected as suggested.

(New L47) L44: "L44 volatile compounds (VOCs)" should be "volatile organic compounds (VOCs)":

Corrected as suggested

L91: "Isolate" should be "strain":

This line has been eliminated." isolate" has been replaced by "strain" whenever possible

(New L135) L100: "Petri" should be "petri dish":

We prefer using Petri as this is the family name of the inventor of this type of plate Julius Richard Petri.

(New L141) L105: WPS should be spelled out:

Done

(New L144) L108: ITS should be spelled out and briefly explained as "internal transcribed spacer", and the nucleotide identity of the sequenced ITS region should be provided to the known sequence:

Done. The meaning of ITS is explained when used for the first time, and the sequence has been deposited and is supplied in the supplementary information.-

Figure 1: Arial with bold font should be used, "P" should be italicized in the legend, and media in Figure 1C should be described with full names and all abbreviations spelled out in the legend:

Corrected as suggested.

(New L144) L134: Degree sign should be checked:

Corrected as suggested.

(New L167) L139: Culture media should be described with full names:

Corrected as suggested.

(New L170) L139 and L142: "PDB" is correct, not "PBD", and media in Figure 1C should be consistent with the rest of the manuscript:

Corrected as suggested. PDB is Potato Dextrose Broth, and PDA is Potato dextrose Agar. In Figure 1C, the test was made in solid agar plates, which is why PDA is used instead of PDB.

Figure 2: it should be indicated which images are from the upper or bottom portion of the plate:

The upper row pictures were taken from below the Erlenmeyer flask and corresponded to pictures of floating mycelial pellets. The lower row corresponds to the pictures of the mycelia after filtering out the culture broth.

Table 1: the thresholds for selecting the number of expressed genes should be described:

In this table, the numbers correspond to all the genes for which transcription was detected

Table 1: "KOG" should be spelled out:

Corrected as suggested.

(New L213) L182-184: the two sentences should be revised for clarity as they are unclear:

Transcription effort was defined as the percentage of the total normalized transcription corresponding to a gene or a group of genes. These results indicate that most of the transcription effort was concentrated on a few genes.

As previously mentioned, we have introduced the concept of "transcription effort" to illustrate the proportion of total transcription within a cell or culture dedicated to producing mRNAs of a specific gene or group of genes. We believe that this straightforward concept is valuable in capturing the actual workload of a cell or culture. The transcription accumulation curves, which have been relocated to the supplementary information as per the suggestion, demonstrate that a significant portion of the cell's transcription is concentrated on a relatively small number of genes. Additionally, it is noteworthy that many of these highly transcribed genes remain functionally uncharacterized. In the revised text, we have tried to explain this concept more precisely.

Figure 4: it can be deleted and replaced with a table showing 10 representative genes with high expression based on TPM values from each transcriptome:

This figure has been moved to the supplementary information along with the others corresponding to the other culture conditions. We include a new table as suggested.

Table 3: it should be deleted:

Done, we explain these data in the text. The table has been moved to the supplementary information

Figure 5: the quality should be improved, and the meaning of each alphabet in the graph should be explained:

Done.

Figures 6, 7, and 8: the quality should be improved:

Done

Comments on the Quality of English Language. This manuscript should be edited by a professional English editing service.

The Mdpi-suggested editorial system has revised English usage

Reviewer 2 Report

In this manuscript Tuber borchii SP1 strain has been isolated and cultured, and its transcriptome has been analyzed under different in vitro culture conditions. The Reviewer considers that the topic has certain importance in the field. However, in Reviewer opinion this work doesn´t have enough scientific merit for publishing in International Journal of Molecular Sciences in its present form. The reliability of the results raises my doubts (no technical repetition of key transcriptomic analysis and no information on deposition of the raw data for the transcriptomes described in this work) and is a very weak point of this work. Some parts of the text (discussion, conclusion, aim of the work) require a thorough revision and correction. Some crucial methodological information is missing. In addition, the Reviewer has major concerns on the manuscript in terms of scientific content and writing (please see specific comments below).

Some key problems that should be addressed by the Authors are discussed below:

1.      Lines 91-95: the scientific goal of this work is poorly formulated. Growth cannot be the goal of the work, nor is the preparation of the transcriptomes.

2.      Lines 98-105: text fragment better suits materials and methods section.

3.      Line 105: What is WPS medium? The abbreviation appears here for the first time. It should be explained.

4.      Figure 1B: It is better way (professional) to provide a scale on the photo, than to write about the diameter of mycelium (figure legend)

5.      Line 135 and 148: What does it mean ‘the best growth’, ‘the highest growth’? The growth can be considered as fast or slow, etc. Please rewrite it and check the whole manuscript in this respect

6.      Figure 4: This is technical information. Should be moved to the supplementary materials together with  similar curves obtained for the transcriptomes made in WPGY, PDB, and MPY.

7.      Lines 176-197: Venn diagram showing e.g. unigenes should be performed for those results also DEG analysis. This would clearly show global changes in the metabolism.

8.      Line 215: ‘It was decided to study the frequency of KOG categories in the expressed genes’ Why? Explain.

9.      Line 219 and 221: ‘was more expressed’??? Than what? Did you mean up regulation?

10.  Figure 5: Provide figure legend for symbols: A-Z in manuscript body not in supplementary materials.

11.  Line 231: Why glycolysis, TCA, glyoxylate cycle, and oxidative phosphorylation were chosen?

12.  Line 240: How the correlation was measured or calculated?

13.  Line 242: above or below?

14.  Line 277: ‘respiratory chain’ should be changed to ‘electron transport chain’ (here and elsewhere)

15.  Line 406-407:  The conclusion is too far reaching. Based on what authors claim that Hsp20 coding gene is relevant in minimal media compared to complex media?

16.  Lines 381-384 and 424-425: in my opinion this disqualifies the work for publication. Bearing in mind that there is no information on the deposition of the raw data, there may be additional  problem with the data quality.

17.  Lines 380-524: the discussion is rather limited to a some kind of explanation or summary of the results. Literature references are missing on 3 pages of key results (only four or five literature references).

18.  Line 538 and 548: Provide more information on WoodPlant. I was not able to find any information on manufacturer’s website.

19.  Lines 613-630: The whole conclusion section should be rewritten. It contains elements of the abstract (should be avoided).

20.  Line 575: I was not able to find ITS accession number in Genbank???

21.  Line 584-604: No information on deposition raw data for the transcriptomes described in this work. According to instructions for authors: ‘New sequence information must be deposited to the appropriate database prior to submission of the manuscript. Accession numbers provided by the database should be included in the submitted manuscript. Manuscripts will not be published until the accession number is provided.’ And ‘New high throughput sequencing (HTS) datasets (RNA-seq, ChIP-Seq, degradome analysis, …) must be deposited either in the GEO database or in the NCBI’s Sequence Read Archive (SRA).’ There is nothing on it. Please, make a deposition of the data to maintain the integrity, transparency and reproducibility of research.

22.  Subsection 2.5-2.7: No information how the results were obtained and the raw data processed.

23.  References section: numerous editorial errors.

24.  Major English revision is required throughout the manuscript. Manuscript contains language and spelling errors and contain substantive errors. Sometimes awkward phrases are used.

Minor corrections:

25.  line 91, 100, 625, Figure 1C.

Therefore, the Reviewer suggests manuscript rejection.

Extensive editing of English language required

Author Response

Referee #2

In this manuscript Tuber borchii SP1 strain has been isolated and cultured, and its transcriptome has been analyzed under different in vitro culture conditions. The Reviewer considers that the topic has certain importance in the field. However, in Reviewer opinion this work doesn't have enough scientific merit for publishing in International Journal of Molecular Sciences in its present form. The reliability of the results raises my doubts (no technical repetition of key transcriptomic analysis and no information on deposition of the raw data for the transcriptomes described in this work) and is a very weak point of this work. Some parts of the text (discussion, conclusion, aim of the work) require a thorough revision and correction. Some crucial methodological information is missing. In addition, the Reviewer has major concerns on the manuscript in terms of scientific content and writing (please see specific comments below).

Some key problems that should be addressed by the Authors are discussed below:

1.- (New L116 to L130) Lines 91-95: the scientific goal of this work is poorly formulated. Growth cannot be the goal of the work, nor is the preparation of the transcriptomes.

We agree with the reviewer's suggestion that the paper's goal can be better explained. The limited availability of Tuber strains amenable to in vitro cultivation under laboratory conditions poses a challenge for molecular studies. Therefore, it is crucial to expand the number of strains that can be cultivated and to establish baseline conditions for future comparisons with experimental strains. This paper addresses these challenges by reporting the isolation and cultivation of a new strain of Tuber (since axenic cultures are impossible for most wild Tuber isolates). We aim to establish a transcriptomic reference landscape that can serve as a proxy for future metabolic studies. We achieve this by analyzing the transcriptomic landscape of Tuber using different culture media, thereby establishing a reference framework. Additionally, the selection of specific metabolic pathways for more in-depth study is motivated by the need to enhance this reference framework further, as we will explain in detail below.

2.- (New L133-L136) Lines 98-105: text fragment better suits materials and methods section.

As mentioned earlier, achieving axenic cultivation of Truffle mycelium is challenging. It is important to note that no cultivable mycelium was obtained from most of our samples. In the rare successful cases, the mycelium was derived from the gleba of mature truffles. We believe that providing this information in this paragraph is relevant and helps to emphasize the difficulties associated with cultivating Truffle mycelium under controlled laboratory conditions..

3.- (New L141) Line 105: What is WPS medium? The abbreviation appears here for the first time. It should be explained:

Corrected as suggested.

4.- Figure 1B: It is better way (professional) to provide a scale on the photo, than to write about the diameter of mycelium (figure legend):

Corrected as suggested.

5.- (New L157)  Line 135 and 148: What does it mean 'the best growth', 'the highest growth'? The growth can be considered as fast or slow, etc. Please rewrite it and check the whole manuscript in this respect:

Corrected as suggested. We were biased by the difficulties in producing biomass in axenic cultures. For that reason, the highest growth was for us the best growth. You are correct in your comment.

6.- Figure 4: This is technical information. Should be moved to the supplementary materials together with similar curves obtained for the transcriptomes made in WPGY, PDB, and MPY:

Corrected as suggested. The figure has been moved to the supplementary information along with the other accumulation curves for the other cultures.

7.- Lines 176-197: Venn diagram showing e.g. unigenes should be performed for those results also DEG analysis. This would clearly show global changes in the metabolism.

Venn diagrams can indeed be informative for illustrating overlapping gene groups. However, in our case, where transcription was detected in most annotated genes in the genome, such representation does not provide valuable information. Additionally, the use of differential expression analysis poses challenges in terms of both purpose and methodology. As we have explained previously, our main objective is to describe the transcriptomic profile of the fungus. Our key questions revolve around identifying the most highly expressed genes and determining the expression levels of genes involved in essential metabolic pathways, crucial for both mycelial growth and commercial interest.

Our approach involves comparing gene expression within a sample rather than between samples. This unique approach allows us to detect significant findings, such as the higher expression of the ADH gene compared to other genes encoding enzymes involved in central metabolism. If we had conducted a comparative analysis, this important observation would have been missed since there are no noticeable expression differences among the analyzed samples. Furthermore, many of the most highly expressed genes are not differentially expressed and would have been overlooked in a standard differential expression analysis. While we acknowledge the importance of DEG analyses in comparative studies, we aim to establish the foundations for such analyses.

Notably, the transcriptomes derived from cultures grown in minimal media (WPG and WPGY) were virtually identical, whereas differences were observed in the transcriptomes derived from complex media. When four independent transcriptomes exhibit shared features, such as the expression levels of central metabolism genes, we envision a transcriptomic landscape that can serve as a practical reference for future comparisons with enhanced quantitative resolution.

8.- Line 215: 'It was decided to study the frequency of KOG categories in the expressed genes' Why? Explain.

We selected the KOG (Clusters of Orthologous Groups) classification because it is one of the three available annotations at the Mycocosm portal, where the genome of the T. borchii reference strain is deposited. Among the available options at the portal, including GO (Gene Ontology) and KEGG (Kyoto Encyclopedia of Genes and Genomes), we opted for KOG to provide a general overview of transcribed genes more simply and straightforwardly. We relied on the annotation provided by KEGG to reconstruct the metabolic pathways of interest. Occasionally, when necessary, we also conducted homology searches to find the pathway's with missing genes. In summary, we chose KOG as our classification scheme for its clarity and simplicity, aligning with the scope of our study.

9.- Line 219 and 221: 'was more expressed'??? Than what? Did you mean up regulation?

We cannot say that. The transcriptional effort allocated to category O is higher in the two cultures grown using minimal media (WPG and WPGY) compared to the culture grown using complex media (MPY and PDB). The transcriptional effort of category O represents the cumulative transcription of all genes within this category, and it would be speculative to assume the upregulation of specific genes or processes. To complement our analysis, we have included a table featuring the top 10 expressed genes in each culture (as suggested by the other referee), and chaperones are among the top ten genes in the WPG and WPGY cultures.

10.- Figure 5: Provide figure legend for symbols: A-Z in manuscript body not in supplementary materials):

Done as required.

11.- Line 231: Why glycolysis, TCA, glyoxylate cycle, and oxidative phosphorylation were chosen?

As previously discussed, our objective was to provide a comprehensive overview of the characteristics of T. borchii SP1 that can serve as a reference for future studies. In line with this goal, we specifically focused on examining the genetic expression in the central metabolic pathways. Our attention was directed towards four key pathways: glycolysis, TCA (tricarboxylic acid cycle), glyoxylate cycle, and oxidative phosphorylation. Through this analysis, we aimed to address certain questions. Firstly, we investigated whether the genes encoding enzymes involved in the production of volatile organic compounds (VOCs) exhibited different expression levels compared to the genes involved in the central metabolism of the fungus.

Additionally, we explored the expression patterns of genes participating in secondary metabolite clusters compared to those involved in the central metabolism. These considerations guided our selection of the glycolysis, TCA, and glyoxylate cycles for examination. Notably, the significant transcriptional values observed for ADH (alcohol dehydrogenase) in all samples prompted us to investigate the Electron Transport Chain as a transcriptional indicator of fungal respiration under these conditions.

12.-  Line 240: How the correlation was measured or calculated?

A simple correlation coefficient was calculated to assess the relationship between the transcription values of the corresponding genes. We want to emphasize that the WPG and WPGY samples exhibited high similarity, distinct from the MPY and PDB samples.

13.-  Line 242: above or below?

We referred to the figure. We eliminate the comment because it can be misleading depending on the final layout of the paper.

14.- Line 277: 'respiratory chain' should be changed to 'electron transport chain' (here and elsewhere):

Done as required.

15.-  Line 406-407:  The conclusion is too far reaching. Based on what authors claim that Hsp20 coding gene is relevant in minimal media compared to complex media?

We agree with you, and we have eliminated the sentence.

16.-  Lines 381-384 and 424-425: in my opinion this disqualifies the work for publication. Bearing in mind that there is no information on the deposition of the raw data, there may be additional  problem with the data quality.

The raw data have been deposited in a public database, and the accession number is provided.

As previously discussed, the main objective of this study is to establish the transcriptome landscape of the Tuber strain under in vitro conditions. We acknowledge that the term "preliminary" may have been misleading, and we apologize for any confusion caused. It is essential to clarify that our results are not provisional but serve as a foundation for future studies exploring gene expression differences under various experimental conditions.

Although we acknowledge that the absence of technical replicates limits the ability to make precise comparisons between transcriptomes under different conditions, the four analyzed transcriptomes can be considered biological replicates reflecting the general behavior of T. borchii in axenic cultures. The WPG and WPGY samples exhibited remarkable similarity, while all four samples displayed consistent transcriptional landscapes. The repetition of these expression profiles supports the notion that these common patterns are a reference for future comparisons. Moreover, these profiles provide valuable insights and suggest new hypotheses for further investigation.

The rationale behind the intra-sample comparison is the assumption that all mRNA molecules are extracted and processed similarly during the RNA sequencing process. This assumption allows us to attribute observed differences in gene expression within a sample to actual variations in transcription levels. We can reasonably infer that these profiles reflect underlying biological processes by comparing the transcription profiles of specific gene sets, such as those involved in the TCA cycle or glycolysis, across different samples obtained from various culture conditions.

Regarding point #7, the case of the ADH gene was discussed to highlight its consistent expression across all four transcriptomes. This finding and similar results in all samples support the hypothesis that T. borchii exhibits a fermentative metabolism under these specific culture conditions.

17.-  Lines 380-524: the discussion is rather limited to some kind of explanation or summary of the results. Literature references are missing on 3 pages of key results (only four or five literature references).

We have rewritten the discussion section incrementing the bibliographical review.

18.-  Line 538 and 548: Provide more information on WoodPlant. I was not able to find any information on manufacturer's website.

Corrected. It was an errata. The correct term is "woody plant"

19.-  Lines 613-630: The whole conclusion section should be rewritten. It contains elements of the abstract (should be avoided).

Done as requested

20.-  Line 575: I was not able to find ITS accession number in Genbank???.

This has been solved, and the accession is already available. It was initially embargoed until the publication of the paper. Additionally, the ITS sequence is provided in the supplementary information.

21.-  Line 584-604: No information on deposition raw data for the transcriptomes described in this work. According to instructions for authors: 'New sequence information must be deposited to the appropriate database prior to submission of the manuscript. Accession numbers provided by the database should be included in the submitted manuscript. Manuscripts will not be published until the accession number is provided.' And 'New high throughput sequencing (HTS) datasets (RNA-seq, ChIP-Seq, degradome analysis, …) must be deposited either in the GEO database or in the NCBI's Sequence Read Archive (SRA).' There is nothing on it. Please, make a deposition of the data to maintain the integrity, transparency and reproducibility of research.

Done, the accession number is GSE233283 

22.-  Subsection 2.5-2.7: No information how the results were obtained and the raw data processed.

The genes involved in the respective pathways were identified using the automatic annotation available at the Mycocosm portal for T. borchii. The KEGG classification system, specifically the map utility, was employed to assign genes to their corresponding pathways. In instances where specific genes were missing or not adequately annotated for a particular step, manual searches were conducted based on gene names or through homology searches.

When multiple genes share the same enzymatic function within a pathway, the transcriptional effort for that particular gene was determined by summing the transcription values of all the relevant genes. However, there was one exception in the case of NADH-dehydrogenase (ETC complex I), where only two genes were annotated corresponding to two subunits of this enzyme complex. The transcription value assigned to complex I was derived from the transcription values of these two genes. It is worth noting that both of these genes exhibited significantly lower transcription values compared to other genes in this pathway.

The raw data used in this analysis are provided in the supplementary information for reference and further exploration.

23.-  References section: numerous editorial errors.

Revised as requested.

24.-  Major English revision is required throughout the manuscript. Manuscript contains language and spelling errors and contain substantive errors. Sometimes awkward phrases are used.

The final manuscript has been submitted to MDPI for English language editing and revision. We have enlisted their services to ensure the highest quality of English usage in the manuscript.

 Minor corrections:

25.-  line 91, 100, 625, Figure 1C.

 Revised and corrected

Round 2

Reviewer 1 Report

The authors have appropriately revised their manuscript in accordance with the comments from the reviewers. 

Therefore, the manuscript is now deemed suitable for publication without further changes.

Reviewer 2 Report

I carefully read the revised manuscript the authors' replies. Authors greatly improved the manuscript. I have only one reservation regarding the results, specifically the sequencing (it turns out that the authors did the sequencing in only one replicate, so the reliability of the results raises some doubts). Therefore, I leave the final decision on the manuscript to the editor. Otherwise, this work is suitable for publication in International Journal of Molecular Sciences in the present form.